# Neural Voice Cloning with a Few Samples

**Sercan Ö. Arık**[*]
sercanarik@baidu.com

**Jitong Chen**[*]
chenjitong01@baidu.com

**Kainan Peng**[*]
pengkainan@baidu.com

**Wei Ping**[*]
pingwei01@baidu.com

**Yanqi Zhou**
yanqiz@baidu.com

Baidu Research
1195 Bordeaux Dr. Sunnyvale, CA 94089

## Abstract

Voice cloning is a highly desired feature for personalized speech interfaces. We introduce a neural voice cloning system that learns to synthesize a person's voice from only a few audio samples. We study two approaches: speaker adaptation and speaker encoding. Speaker adaptation is based on fine-tuning a multi-speaker generative model. Speaker encoding is based on training a separate model to directly infer a new speaker embedding, which will be applied to a multi-speaker generative model. In terms of naturalness of the speech and similarity to the original speaker, both approaches can achieve good performance, even with a few cloning audios. [2] While speaker adaptation can achieve slightly better naturalness and similarity, cloning time and required memory for the speaker encoding approach are significantly less, making it more favorable for low-resource deployment.

## 1 Introduction

Generative models based on deep neural networks have been successfully applied to many domains such as image generation [e.g., Oord et al., 2016b, Karras et al., 2017], speech synthesis [e.g., Oord et al., 2016a, Arik et al., 2017a, Wang et al., 2017], and language modeling [e.g., Jozefowicz et al., 2016]. Deep neural networks are capable of modeling complex data distributions and can be further conditioned on external inputs to control the content and style of generated samples.

In speech synthesis, generative models can be conditioned on text and speaker identity [e.g., Arik et al., 2017b]. While text carries linguistic information and controls the content of the generated speech, speaker identity captures characteristics such as pitch, speech rate and accent. One approach for multi-speaker speech synthesis is to jointly train a generative model and speaker embeddings on triplets of text, audio and speaker identity [e.g., Ping et al., 2018]. The idea is to encode the speaker-dependent information with low-dimensional embeddings, while sharing the majority of the model parameters across all speakers. One limitation of such methods is that they can only generate speech for observed speakers during training. An intriguing task is to learn the voice of an unseen speaker from a few speech samples, a.k.a. *voice cloning*, which corresponds to few-shot generative modeling of speech conditioned on the speaker identity. While a generative model can be trained from scratch with a large amount of audio samples [3], we focus on voice cloning of a new speaker with a few minutes or even few seconds data. It is challenging as the model has to learn the speaker characteristics from very limited amount of data, and still generalize to unseen texts.

---

[*]Equal contribution

[2]Cloned audio samples can be found in `https://audiodemos.github.io`

[3]A single speaker model can require ∼20 hours of training data [e.g., Arik et al., 2017a, Wang et al., 2017], while a multi-speaker model for 108 speakers [Arik et al., 2017b] requires about ∼20 minutes data per speaker.

In this paper, we investigate voice cloning in sequence-to-sequence neural speech synthesis systems [Ping et al., 2018]. Our contributions are the following:

1. We demonstrate and analyze the strength of speaker adaption approaches for voice cloning, based on fine-tuning a pre-trained multi-speaker model for an unseen speaker using a few samples.

2. We propose a novel speaker encoding approach, which provides comparable naturalness and similarity in subjective evaluations while yielding significantly less cloning time and computational resource requirements.

3. We propose automated evaluation methods for voice cloning based on neural speaker classification and speaker verification.

4. We demonstrate voice morphing for gender and accent transformation via embedding manipulations.

## 2  Related Work

Our work builds upon the state-of-the-art in neural speech synthesis and few-shot generative modeling.

**Neural speech synthesis:**  Recently, there is a surge of interest in speech synthesis with neural networks, including Deep Voice 1 [Arik et al., 2017a], Deep Voice 2 [Arik et al., 2017b], Deep Voice 3 [Ping et al., 2018], WaveNet [Oord et al., 2016a], SampleRNN [Mehri et al., 2016], Char2Wav [Sotelo et al., 2017], Tacotron [Wang et al., 2017] and VoiceLoop [Taigman et al., 2018]. Among these methods, sequence-to-sequence models [Ping et al., 2018, Wang et al., 2017, Sotelo et al., 2017] with attention mechanism have much simpler pipeline and can produce more natural speech [e.g., Shen et al., 2017]. In this work, we use Deep Voice 3 as the baseline multi-speaker model, because of its simple convolutional architecture and high efficiency for training and fast model adaptation. It should be noted that our techniques can be seamlessly applied to other neural speech synthesis models.

**Few-shot generative modeling:**  Humans can learn new generative tasks from only a few examples, which motivates research on few-shot generative models. Early studies mostly focus on Bayesian methods. For example, hierarchical Bayesian models are used to exploit compositionality and causality for few-shot generation of characters [Lake et al., 2013, 2015] and words in speech [Lake et al., 2014]. Recently, deep neural networks achieve great successes in few-shot density estimation and conditional image generation [e.g., Rezende et al., 2016, Reed et al., 2017, Azadi et al., 2017], because of the great potential for composition in their learned representation. In this work, we investigate few-shot generative modeling of speech conditioned on a particular speaker. We train a separate speaker encoding network to directly predict the parameters of multi-speaker generative model by only taking unsubscribed audio samples as inputs.

**Speaker-dependent speech processing:**  Speaker-dependent modeling has been widely studied for automatic speech recognition (ASR), with the goal of improving the performance by exploiting speaker characteristics. In particular, there are two groups of methods in neural ASR, in alignment with our two voice cloning approaches. The first group is speaker adaptation for the whole-model [Yu et al., 2013], a portion of the model [Miao and Metze, 2015, Cui et al., 2017], or merely to a speaker embedding [Abdel-Hamid and Jiang, 2013, Xue et al., 2014]. Speaker adaptation for voice cloning is in the same vein as these approaches, but differences arise when text-to-speech vs. speech-to-text are considered [Yamagishi et al., 2009]. The second group is based on training ASR models jointly with embeddings. Extraction of the embeddings can be based on *i-vectors* [Miao et al., 2015], or bottleneck layers of neural networks trained with a classification loss [Li and Wu, 2015]. Although the general idea of speaker encoding is also based on extracting the embeddings directly, as a major distinction, our speaker encoder models are trained with an objective function that is directly related to speech synthesis. Lastly, speaker-dependent modeling is essential for multi-speaker speech synthesis. Using *i-vectors* to represent speaker-dependent characteristics is one approach [Wu et al., 2015], however, they have the limitation of being separately trained, with an objective that is not directly related to speech synthesis. Also they may not be accurately extracted with small amount of audio [Miao et al., 2015]. Another approach for multi-speaker speech synthesis is using trainable speaker embeddings [Arik et al., 2017b], which are randomly initialized and jointly optimized from a generative loss function.

**Voice conversion:** A closely related task of voice cloning is voice conversion. The goal of voice conversion is to modify an utterance from source speaker to make it sound like the target speaker, while keeping the linguistic contents unchanged. Unlike voice cloning, voice conversion systems do not need to generalize to unseen texts. One common approach is dynamic frequency warping, to align spectra of different speakers. Agiomyrgiannakis and Roupakia [2016] proposes a dynamic programming algorithm that simultaneously estimates the optimal frequency warping and weighting transform while matching source and target speakers using a matching-minimization algorithm. Wu et al. [2016] uses a spectral conversion approach integrated with the locally linear embeddings for manifold learning. There are also approaches to model spectral conversion using neural networks [Desai et al., 2010, Chen et al., 2014, Hwang et al., 2015]. Those models are typically trained with a large amount of audio pairs of target and source speakers.

## 3    From Multi-Speaker Generative Modeling to Voice Cloning

We consider a multi-speaker generative model, $f(\mathbf{t}_{i,j}, s_i; W, \mathbf{e}_{s_i})$, which takes a text $\mathbf{t}_{i,j}$ and a speaker identity $s_i$. The trainable parameters in the model is parameterized by $W$, and $\mathbf{e}_{s_i}$. The latter denotes the trainable speaker embedding corresponding to $s_i$. Both $W$ and $\mathbf{e}_{s_i}$ are optimized by minimizing a loss function $L$ that penalizes the difference between generated and ground-truth audios (e.g. a regression loss for spectrograms):

$$\min_{W,\mathbf{e}} \mathbb{E}_{\substack{s_i \sim \mathcal{S}, \\ (\mathbf{t}_{i,j}, \mathbf{a}_{i,j}) \sim \mathcal{T}_{s_i}}} \left\{ L\left(f(\mathbf{t}_{i,j}, s_i; W, \mathbf{e}_{s_i}), \mathbf{a}_{i,j}\right) \right\} \tag{1}$$

where $\mathcal{S}$ is a set of speakers, $\mathcal{T}_{s_i}$ is a training set of text-audio pairs for speaker $s_i$, and $\mathbf{a}_{i,j}$ is the ground-truth audio for $\mathbf{t}_{i,j}$ of speaker $s_i$. The expectation is estimated over text-audio pairs of all training speakers. We use $\widehat{W}$ and $\widehat{\mathbf{e}}$ to denote the trained parameters and embeddings. Speaker embeddings have been shown to effectively capture speaker characteristics with low-dimensional vectors [Arik et al., 2017b, Ping et al., 2018]. Despite training with only a generative loss, discriminative properties (e.g. gender and accent) are observed in the speaker embedding space [Arik et al., 2017b].

For voice cloning, we extract the speaker characteristics for an unseen speaker $s_k$ from a set of cloning audios $\mathcal{A}_{s_k}$, and generate an audio given any text for that speaker. The two performance metrics for the generated audio are speech naturalness and speaker similarity (i.e., whether the generated audio sounds like it is pronounced by the target speaker). The two approaches for neural voice cloning are summarized in Fig. 1 and explained in the following sections.

### 3.1    Speaker adaptation

The idea of speaker adaptation is to fine-tune a trained multi-speaker model for an unseen speaker using a few audio-text pairs. Fine-tuning can be applied to either the speaker embedding [Taigman et al., 2018] or the whole model. For embedding-only adaptation, we have the following objective:

$$\min_{\mathbf{e}_{s_k}} \mathbb{E}_{(\mathbf{t}_{k,j}, \mathbf{a}_{k,j}) \sim \mathcal{T}_{s_k}} \left\{ L\left(f(\mathbf{t}_{k,j}, s_k; \widehat{W}, \mathbf{e}_{s_k}), \mathbf{a}_{k,j}\right) \right\}, \tag{2}$$

where $\mathcal{T}_{s_k}$ is a set of text-audio pairs for the target speaker $s_k$. For whole model adaptation, we have the following objective:

$$\min_{W,\mathbf{e}_{s_k}} \mathbb{E}_{(\mathbf{t}_{k,j}, \mathbf{a}_{k,j}) \sim \mathcal{T}_{s_k}} \left\{ L\left(f(\mathbf{t}_{k,j}, s_k; W, \mathbf{e}_{s_k}), \mathbf{a}_{k,j}\right) \right\}. \tag{3}$$

Although the entire model provides more degrees of freedom for speaker adaptation, its optimization is challenging for small amount of cloning data. Early stopping is required to avoid overfitting.

### 3.2    Speaker encoding

We propose a speaker encoding method to directly estimate the speaker embedding from audio samples of an unseen speaker. Such a model does not require any fine-tuning during voice cloning. Thus, the same model can be used for all unseen speakers. The speaker encoder, $g(\mathcal{A}_{s_k}; \mathbf{\Theta})$, takes a set of cloning audio samples $\mathcal{A}_{s_k}$ and estimates $\mathbf{e}_{s_k}$ for speaker $s_k$. The model is parametrized by $\mathbf{\Theta}$. Ideally, the speaker encoder can be jointly trained with the multi-speaker generative model from scratch, with a loss function defined for the generated audio:

$$\min_{W,\mathbf{\Theta}} \mathbb{E}_{\substack{s_i \sim \mathcal{S}, \\ (\mathbf{t}_{i,j}, \mathbf{a}_{i,j}) \sim \mathcal{T}_{s_i}}} \left\{ L\left(f(\mathbf{t}_{i,j}, s_i; W, g(\mathcal{A}_{s_i}; \mathbf{\Theta})), \mathbf{a}_{i,j}\right) \right\}. \tag{4}$$

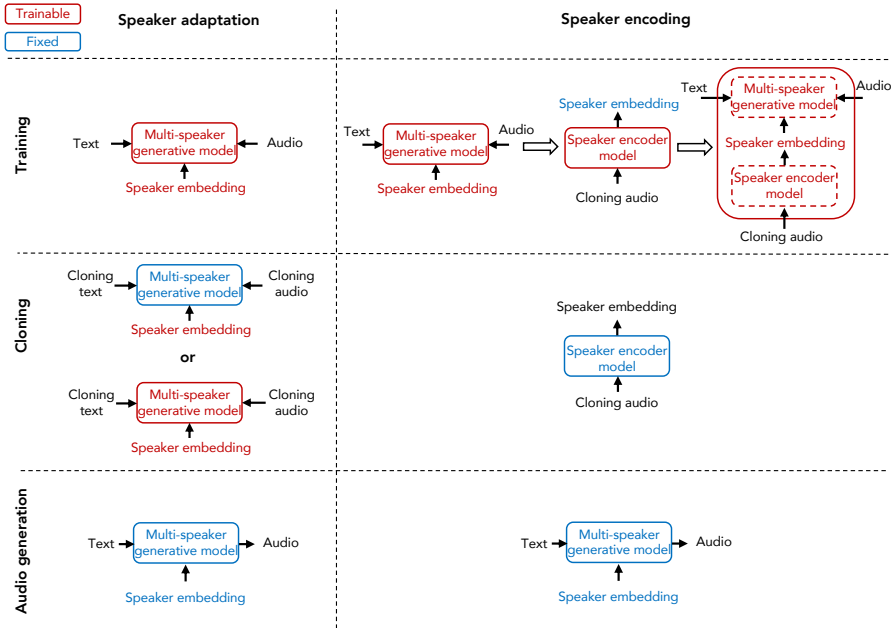

Figure 1: Illustration of speaker adaptation and speaker encoding approaches for voice cloning.

Note that the speaker encoder is trained with the speakers for the multi-speaker generative model. During training, a set of cloning audio samples $\mathcal{A}_{s_i}$ are randomly sampled for training speaker $s_i$. During inference, audio samples from the target speaker $s_k$, $\mathcal{A}_{s_k}$, are used to compute $g(\mathcal{A}_{s_k}; \Theta)$. We observed optimization challenges with joint training from scratch: the speaker encoder tends to estimate an average voice to minimize the overall generative loss. One possible solution is to introduce discriminative loss functions for intermediate embeddings[4] or generated audios[5]. In our case, however, such approaches only slightly improve speaker differences. Instead, we propose a separate training procedure for speaker encoder. Speaker embeddings $\widehat{\mathbf{e}}_{\mathbf{s_i}}$ are extracted from a trained multi-speaker generative model $f(\mathbf{t}_{i,j}, s_i; W, \mathbf{e}_{s_i})$. Then, the speaker encoder $g(\mathcal{A}_{s_k}; \Theta)$ is trained with an L1 loss to predict the embeddings from sampled cloning audios:

$$\min_{\Theta} \mathbb{E}_{s_i \sim \mathcal{S}} \left\{ |g(\mathcal{A}_{s_i}; \Theta) - \widehat{\mathbf{e}}_{\mathbf{s_i}})| \right\}. \tag{5}$$

Eventually, entire model can be jointly fine-tuned following Eq. 4, with pre-trained $\widehat{W}$ and $\widehat{\Theta}$ as the initialization. Fine-tuning encourages the generative model to compensate for embedding estimation errors, and may reduce attention problems. However, generative loss still dominates learning and speaker differences in generated audios may be slightly reduced as well (see Section 4.3 for details).

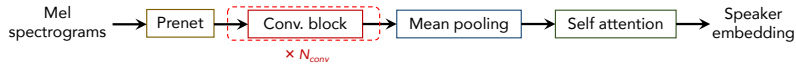

Figure 2: Speaker encoder architecture. See Appendix A for details.

For speaker encoder $g(\mathcal{A}_{s_k}; \Theta)$, we propose a neural network architecture comprising three parts (as shown in Fig. 2):

(i) *Spectral processing*: We input mel-spectrograms of cloning audio samples to prenet, which contains fully-connected layers with exponential linear unit for feature transformation.

(ii) *Temporal processing*: To utilize long-term context, we use convolutional layers with gated linear unit and residual connections, average pooling is applied to summarize the whole utterance.

(iii) *Cloning sample attention*: Considering that different cloning audios contain different amount of speaker information, we use a multi-head self-attention mechanism [Vaswani et al., 2017] to compute the weights for different audios and get aggregated embeddings.

### 3.3 Discriminative models for evaluation

Besides human evaluations, we propose two evaluation methods using discriminative models for voice cloning performance.

#### 3.3.1 Speaker classification

Speaker classifier determines which speaker an audio sample belongs to. For voice cloning evaluation, a speaker classifier is trained with the set of speakers used for cloning. High-quality voice cloning would result in high classification accuracy. The architecture is composed of similar spectral and temporal processing layers in Fig. 6 and an additional embedding layer before the softmax function.

#### 3.3.2 Speaker verification

Speaker verification is the task of authenticating the claimed identity of a speaker, based on a test audio and enrolled audios from the speaker. In particular, it performs binary classification to identify whether the test audio and enrolled audios are from the same speaker [e.g., Snyder et al., 2016]. We consider an end-to-end text-independent speaker verification model [Snyder et al., 2016] (see Appendix C for more details of model architecture). The speaker verification model can be trained on a multi-speaker dataset, and then used to verify if the cloned audio and the ground-truth audio are from the same speaker. Unlike the speaker classification approach, speaker verification model does not require training with the audios from the target speaker for cloning, hence it can be used for unseen speakers with a few samples. As the quantitative performance metric, the equal error-rate (EER) [6] can be used to measure how close the cloned audios are to the ground truth audios.

## 4 Experiments

### 4.1 Datasets

In our first set of experiments (Sections 4.3 and 4.4), the multi-speaker generative model and speaker encoder are trained using LibriSpeech dataset [Panayotov et al., 2015], which contains audios (16 KHz) for 2484 speakers, totalling 820 hours. LibriSpeech is a dataset for automatic speech recognition, and its audio quality is lower compared to speech synthesis datasets.[7] Voice cloning is performed on VCTK dataset [Veaux et al., 2017]. VCTK consists of audios sampled at 48 KHz for 108 native speakers of English with various accents. To be consistent with LibriSpeech dataset, VCTK audios are downsampled to 16 KHz. For a chosen speaker, a few cloning audios are randomly sampled for each experiment. The sentences presented in Appendix B are used to generate audios for evaluation. In our second set of experiments (Section 4.5), we aim to investigate the impact of the training dataset. We split the VCTK dataset for training and testing: 84 speakers are used for training the multi-speaker model, 8 speakers for validation, and 16 speakers for cloning.

### 4.2 Model specifications

Our multi-speaker generative model is based on the convolutional sequence-to-sequence architecture proposed in Ping et al. [2018], with similar hyperparameters and Griffin-Lim vocoder. To get better performance, we increase the time-resolution by reducing the hop length and window size parameters to 300 and 1200, and add a quadratic loss term to penalize large amplitude components superlinearly. For speaker adaptation experiments, we reduce the embedding dimensionality to 128, as it yields less overfitting problems. Overall, the baseline multi-speaker generative model has around 25M trainable parameters when trained for the LibriSpeech dataset. For the second set of experiments, hyperparameters of the VCTK model is used from Ping et al. [2018] to train a multi-speaker model for the 84 speakers of VCTK, with Griffin-Lim vocoder.

We train speaker encoders for different number of cloning audios separately. Initially, cloning audios are converted to log-mel spectrograms with 80 frequency bands, with a hop length of 400, a window size of 1600. Log-mel spectrograms are fed to spectral processing layers, which are composed of 2-layer prenet of size 128. Then, temporal processing is applied with two 1-D convolutional layers with a filter width of 12. Finally, multi-head attention is applied with 2 heads and a unit size of 128 for keys, queries and values. The final embedding size is 512. Validation set consists 25 held-out speakers. A batch size of 64 is used, with an initial learning rate of 0.0006 with annealing rate of 0.6 applied every 8000 iterations. Mean absolute error for the validation set is shown in Fig. 11 in Appendix D. More cloning audios leads to more accurate speaker embedding estimation, especially with the attention mechanism (see Appendix D for more details about the learned attention coefficients).

We train a speaker classifier using VCTK dataset to classify which of the 108 speakers an audio sample belongs to. Speaker classifier has a fully-connected layer of size 256, 6 convolutional layers with 256 filters of width 4, and a final embedding layer of size 32. The model achieves 100% accuracy for validation set of size 512.

We train a speaker verification model using LibriSpeech dataset. Validation sets consists 50 held-out speakers from Librispeech. EERs are estimated by randomly pairing up utterances from the same or different speakers (50% for each case) in test set. We perform 40960 trials for each test set. We describe the details of speaker verification model in Appendix C.

### 4.3 Voice cloning performance

|  | Speaker adaptation | | Speaker encoding | |
|---|---|---|---|---|
| **Approaches** | Embedding-only | Whole-model | Without fine-tuning | With fine-tuning |
| **Data** | Text and audio | | Audio | |
| **Cloning time** | $\sim 8$ hours | $\sim 0.5 - 5$ mins | $\sim 1.5 - 3.5$ secs | $\sim 1.5 - 3.5$ secs |
| **Inference time** | $\sim 0.4 - 0.6$ secs | | | |
| **Parameters per speaker** | 128 | $\sim 25$ million | 512 | 512 |

Table 1: Comparison of speaker adaptation and speaker encoding approaches.

For speaker adaptation approach, we pick the optimal number of iterations using speaker classification accuracy. For speaker encoding, we consider voice cloning with and without joint fine-tuning of the speaker encoder and multi-speaker generative model.[8] Table 1 summarizes the approaches and lists the requirements for training, data, cloning time and memory footprint.

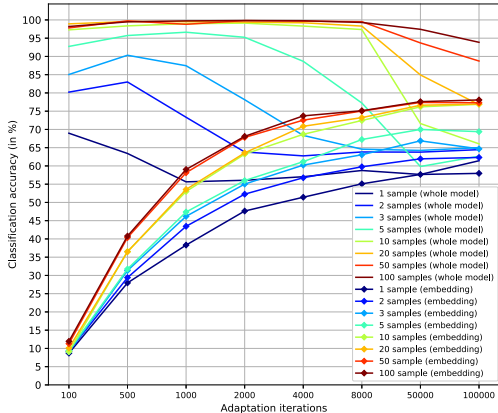

Figure 3: Performance of whole-model adaptation and speaker embedding adaptation for voice cloning in terms of speaker classification accuracy for 108 VCTK speakers.

For speaker adaptation, Fig. 3 shows the speaker classification accuracy vs. the number of iterations. For both, the classification accuracy significantly increases with more samples, up to ten samples. In the low sample count regime, adapting the speaker embedding is less likely to overfit the samples

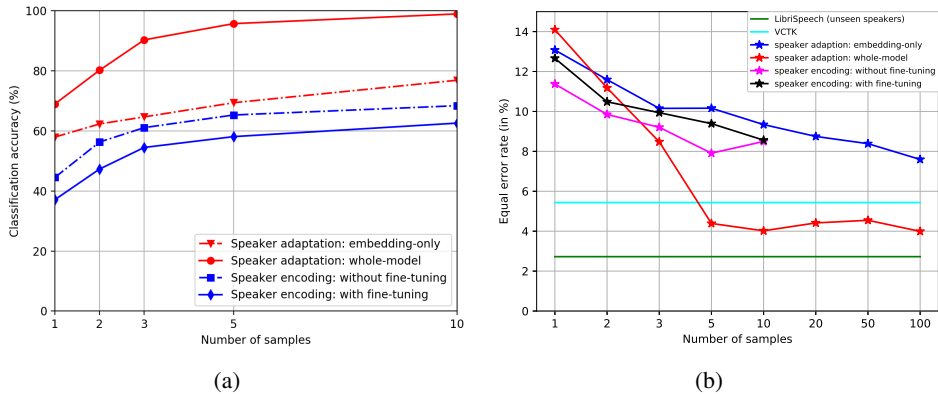

Figure 4: (a) Speaker classification accuracy with different numbers of cloning samples. (b) EER (using 5 enrollment audios) for different numbers of cloning samples. LibriSpeech (unseen speakers) and VCTK represent EERs estimated from random pairing of utterances from ground-truth datasets.

than adapting the whole model. The two methods also require different numbers of iterations to converge. Compared to whole-model adaptation (which converges around 1000 iterations for even 100 cloning audio samples), embedding adaptation takes significantly more iterations to converge, thus it results in much longer cloning time.

Figs. 4a and 4b show the classification accuracy and EER, obtained by speaker classification and speaker verification models. Both speaker adaptation and speaker encoding benefit from more cloning audios. When the number of cloning audio samples exceed five, whole-model adaptation outperforms other techniques. Speaker encoding yields a lower classification accuracy compared to embedding adaptation, but they achieve a similar speaker verification performance.

Besides evaluations by discriminative models, we conduct subject tests on Amazon Mechanical Turk framework. For assessment of the naturalness, we use the 5-scale mean opinion score (MOS). For assessment of how similar the generated audios are to the ground-truth audios from target speakers, we use the 4-scale similarity score with the question and categories in [Wester et al., 2016].[9] Tables 2 and 3 show the results of human evaluations. Higher number of cloning audios improve both metrics. The improvement is more significant for whole model adaptation, due to the more degrees of freedom provided for an unseen speaker. Indeed, for high sample counts, the naturalness significantly exceeds the baseline model, due to the dominance of better quality adaptation samples over training data. Speaker encoding achieves naturalness similar or better than the baseline model. The naturalness is even further improved with fine-tuning since it allows the generative model to learn how to compensate for the errors of the speaker encoder. Similarity scores slightly improve with higher sample counts for speaker encoding, and match the scores for speaker embedding adaptation.

| Approach | Sample count | | | | |
|---|---|---|---|---|---|
| | 1 | 2 | 3 | 5 | 10 |
| Ground-truth (16 KHz sampling rate) | 4.66±0.06 | | | | |
| Multi-speaker generative model | 2.61±0.10 | | | | |
| Speaker adaptation (embedding-only) | 2.27±0.10 | 2.38±0.10 | 2.43±0.10 | 2.46±0.09 | 2.67±0.10 |
| Speaker adaptation (whole-model) | 2.32±0.10 | 2.87±0.09 | 2.98±0.11 | 2.67±0.11 | 3.16±0.09 |
| Speaker encoding (without fine-tuning) | 2.76±0.10 | 2.76±0.09 | 2.78±0.10 | 2.75±0.10 | 2.79±0.10 |
| Speaker encoding (with fine-tuning) | 2.93±0.10 | 3.02±0.11 | 2.97±0.1 | 2.93±0.10 | 2.99±0.12 |

Table 2: Mean Opinion Score (MOS) evaluations for naturalness with 95% confidence intervals (training with LibriSpeech speakers and cloning with 108 VCTK speakers).

| Approach | Sample count | | | | |
|---|---|---|---|---|---|
| | 1 | 2 | 3 | 5 | 10 |
| Ground-truth (same speaker) | 3.91±0.03 | | | | |
| Ground-truth (different speakers) | 1.52±0.09 | | | | |
| Speaker adaptation (embedding-only) | 2.66±0.09 | 2.64±0.09 | 2.71±0.09 | 2.78±0.10 | 2.95±0.09 |
| Speaker adaptation (whole-model) | 2.59±0.09 | 2.95±0.09 | 3.01±0.10 | 3.07±0.08 | 3.16±0.08 |
| Speaker encoding (without fine-tuning) | 2.48±0.10 | 2.73±0.10 | 2.70±0.11 | 2.81±0.10 | 2.85±0.10 |
| Speaker encoding (with fine-tuning) | 2.59±0.12 | 2.67±0.12 | 2.73±0.13 | 2.77±0.12 | 2.77±0.11 |

Table 3: Similarity score evaluations with 95% confidence intervals (training with LibriSpeech speakers and cloning with 108 VCTK speakers).

## 4.4 Voice morphing via embedding manipulation

As shown in Fig. 5, speaker encoder maps speakers into a meaningful latent space. Inspired by word embedding manipulation (e.g. to demonstrate the existence of simple algebraic operations as *king - queen = male - female*), we apply algebraic operations to inferred embeddings to transform their speech characteristics. To transform gender, we estimate the averaged speaker embeddings for each gender, and add their difference to a particular speaker. For example, *BritishMale + AveragedFemale - AveragedMale* yields a British female speaker. Similarly, we consider region of accent transformation via *BritishMale + AveragedAmerican - AveragedBritish* to obtain an American male speaker. Our results demonstrate high quality audios with specific gender and accent characteristics.[10]

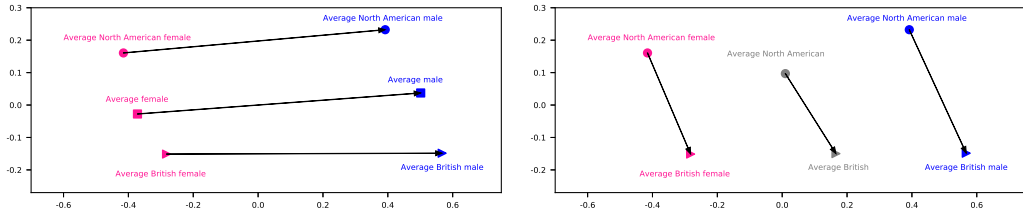

Figure 5: Visualization of estimated speaker embeddings by speaker encoder. The first two principal components of speaker embeddings (averaged across 5 samples for each speaker). Only British and North American regional accents are shown as they constitute the majority of the labeled speakers in the VCTK dataset. Please see Appendix E for more detailed analysis.

## 4.5 Impact of training dataset

| Approach | Sample count | | | | |
|---|---|---|---|---|---|
| | 1 | 5 | 10 | 20 | 100 |
| Speaker adaptation (embedding-only) | 3.01±0.11 | - | 3.13±0.11 | - | 3.13±0.11 |
| Speaker adaptation (whole-model) | 2.34±0.13 | 2.99±0.10 | 3.07±0.09 | 3.40±0.10 | 3.38±0.09 |

Table 4: Mean Opinion Score (MOS) evaluations for naturalness with 95% confidence intervals (training with 84 VCTK speakers and cloning with 16 VCTK speakers).

| Approach | Sample count | | | | |
|---|---|---|---|---|---|
| | 1 | 5 | 10 | 20 | 100 |
| Speaker adaptation (embedding-only) | 2.42±0.13 | - | 2.37±0.13 | - | 2.37±0.12 |
| Speaker adaptation (whole-model) | 2.55±0.11 | 2.93±0.11 | 2.95±0.10 | 3.01±0.10 | 3.14±0.10 |

Table 5: Similarity score evaluations with 95% confidence intervals (training with 84 VCTK speakers and cloning with 16 VCTK speakers).

To evaluate the impact of the dataset, we consider training with a subset of the VCTK containing 84 speakers, and cloning on another 16 speakers. Tables 4 and 5 present the human evaluations

for speaker adaptation.[11] Speaker verification results are given in Appendix C. One the one hand, compared to LibriSpeech, cleaner VCTK data improves the multi-speaker generative model, leading to better whole-model adaptation results. On the other hand, embedding-only adaptation significantly underperforms whole-model adaptation due to the limited speaker diversity in VCTK dataset.

## 5   Conclusions

We study two approaches for neural voice cloning: speaker adaptation and speaker encoding. We demonstrate that both approaches can achieve good cloning quality even with only a few cloning audios. For naturalness, we show that both speaker adaptation and speaker encoding can achieve an MOS similar to the baseline multi-speaker generative model. Thus, the proposed techniques can potentially be improved with better multi-speaker models in the future (such as replacing Griffin-Lim with WaveNet vocoder). For similarity, we demonstrate that both approaches benefit from a larger number of cloning audios. The performance gap between whole-model and embedding-only adaptation indicates that some discriminative speaker information still exists in the generative model besides speaker embeddings. The benefit of compact representation via embeddings is fast cloning and small footprint per speaker. We observe drawbacks of training the multi-speaker generative model using a speech recognition dataset with low-quality audios and limited speaker diversity. Improvements in the quality of dataset would result in higher naturalness. We expect our techniques to benefit significantly from a large-scale and high-quality multi-speaker dataset.

## Footnotes

[4]We have experimented classification loss by mapping the embeddings to labels via a softmax layer.

[5]We have experimented integrating a pre-trained classifier to encourage discrimination in generated audios.

[6]EER is the point when the false acceptance rate and false rejection rate are equal.

[7]We designed a segmentation and denoising pipeline to process LibriSpeech, as in Ping et al. [2018].

[8]The learning rate and annealing parameters are optimized for joint fine-tuning.

[9]We conduct each evaluation independently, so the cloned audios of two different models are not directly compared during rating. Multiple votes on the same sample are aggregated by a majority voting rule.

[10]https://audiodemos.github.io/

[11]The speaker encoder models generalize poorly for unseen speakers due to limited training speakers.

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
