[Supplementary Material · appendices.pdf]

# Appendices

## A   Detailed speaker encoder architecture

Speaker embedding

Figure 6: Speaker encoder architecture with intermediate state dimensions. ($batch$: batch size, $N_{samples}$: number of cloning audio samples $|A_{s_k}|$, $T$: number of mel spectrograms timeframes, $F_{mel}$: number of mel frequency channels, $F_{mapped}$: number of frequency channels after prenet, $d_{embedding}$: speaker embedding dimension). Multiplication operation at the last layer represents inner product along the dimension of cloning samples.

## B   Voice cloning test sentences

```
Prosecutors have opened a massive investigation/into allegations of/fixing games/and illegal betting%.
Different telescope designs/perform differently%and have different strengths/and weaknesses%.
We can continue to strengthen the education of good lawyers%.
Feedback must be timely/and accurate/throughout the project%.
Humans also judge distance/by using the relative sizes of objects%.
Churches should not encourage it%or make it look harmless%.
Learn about/setting up/wireless network configuration%.
You can eat them fresh cooked%or fermented%.
If this is true%then those/who tend to think creatively%really are somehow different%.
She will likely jump for joy%and want to skip straight to the honeymoon%.
The sugar syrup/should create very fine strands of sugar%that drape over the handles%.
But really in the grand scheme of things%this information is insignificant%.
I let the positive/overrule the negative%.
He wiped his brow/with his forearm%.
Instead of fixing it%they give it a nickname%.
About half the people%who are infected%also lose weight%.
The second half of the book%focuses on argument/and essay writing%.
We have the means/to help ourselves%.
The large items/are put into containers/for disposal%.
He loves to/watch me/drink this stuff%.
Still%it is an odd fashion choice%.
Funding is always an issue/after the fact%.
Let us/encourage each other%.
```

Figure 7: The sentences used to generate test samples for the voice cloning models. The white space characters / and % follow the same definition as in Ping et al. [2018].

# C  Speaker verification model

Given a set of (e.g., 1∼5) enrollment audios [12] and a test audio, speaker verification model performs a binary classification and tells whether the enrollment and test audios are from the same speaker. Although using other speaker verification models [e.g., Snyder et al., 2016] would also suffice, we choose to create our own speaker verification models using convolutional-recurrent architecture [Amodei et al., 2016]. We note that our equal-error-rate results on test set of unseen speakers are on par with the state-of-the-art speaker verification models. The architecture of our model is illustrated in Figure 8. We compute mel-scaled spectrogram of enrollment audios and test audio after resampling the input to a constant sampling frequency. Then, we apply two-dimensional convolutional layers convolving over both time and frequency bands, with batch normalization and ReLU non-linearity after each convolution layer. The output of last convolution layer is feed into a recurrent layer (GRU). We then mean-pool over time (and enrollment audios if there are many), then apply a fully connected layer to obtain the speaker encodings for both enrollment audios and test audio. We use the probabilistic linear discriminant analysis (PLDA) for scoring the similarity between the two encodings [Prince and Elder, 2007, Snyder et al., 2016]. The PLDA score [Snyder et al., 2016] is defined as,

$$s(\boldsymbol{x}, \boldsymbol{y}) = w \cdot \boldsymbol{x}^\top \boldsymbol{y} - \boldsymbol{x}^\top S \boldsymbol{x} - \boldsymbol{y}^\top S \boldsymbol{y} + b \tag{6}$$

where $\boldsymbol{x}$ and $\boldsymbol{y}$ are speaker encodings of enrollment and test audios respectively after fully-connected layer, $w$ and $b$ are scalar parameters, and $S$ is a symmetric matrix. Then, $s(\boldsymbol{x}, \boldsymbol{y})$ is feed into a sigmoid unit to obtain the probability that they are from the same speaker. The model is trained using cross-entropy loss. Table 6 lists hyperparameters of speaker verification model for LibriSpeech dataset.

In addition to speaker verification test results presented in main text (Figure 4b), we also include the result using 1 enrollment audio when the multi-speaker generative model is trained on LibriSpeech. When multi-speaker generative model is trained on VCTK, the results are in Figure 10. It should be noted that, the EER on cloned audios could be potentially better than on ground truth VCTK, because the speaker verification model is trained on LibriSpeech dataset.

Figure 8: Architecture of speaker verification model.

| Parameter | |
|---|---|
| Audio resampling freq. | 16 KHz |
| Bands of Mel-spectrogram | 80 |
| Hop length | 400 |
| Convolution layers, channels, filter, strides | $1, 64, 20 \times 5, 8 \times 2$ |
| Recurrent layer size | 128 |
| Fully connected size | 128 |
| Dropout probability | 0.9 |
| Learning Rate | $10^{-3}$ |
| Max gradient norm | 100 |
| Gradient clipping max. value | 5 |

Table 6: Hyperparameters of speaker verification model for LibriSpeech dataset.

Figure 9: Speaker verification EER (using 1 enrollment audio) vs. number of cloning audio samples. Multi-speaker generative model and speaker verification model are trained using LibriSpeech dataset. Voice cloing is performed using VCTK dataset.

(a)                                        (b)

Figure 10: Speaker verification EER using (a) 1 enrollment audio (b) 5 enrollment audios vs. number of cloning audio samples. Multi-speaker generative model is trained on a subset of VCTK dataset including 84 speakers, and voice cloning is performed on other 16 speakers. Speaker verification model is trained using the LibriSpeech dataset.

# D    Implications of attention

For a trained speaker encoder model, Fig. 12 exemplifies attention distributions for different audio lengths. The attention mechanism can yield highly non-uniformly distributed coefficients while combining the information in different cloning samples, and especially assigns higher coefficients to longer audios, as intuitively expected due to the potential more information content in them.

Figure 11: Mean absolute error in embedding estimation vs. the number of cloning audios for a validation set of 25 speakers, shown with the attention mechanism and without attention mechanism (by simply averaging).

Figure 12: Inferred attention coefficients for the speaker encoder model with $N_{samples}$ = 5 vs. lengths of the cloning audio samples. The dashed line corresponds to the case of averaging all cloning audio samples.

# E  Speaker embedding space learned by the encoder

To analyze the speaker embedding space learned by the trained speaker encoders, we apply principal component analysis to the space of inferred embeddings and consider their ground truth labels for gender and region of accent from the VCTK dataset. Fig. 13 shows visualization of the first two principal components. We observe that speaker encoder maps the cloning audios to a latent space with highly meaningful discriminative patterns. In particular for gender, a one dimensional linear transformation from the learned speaker embeddings can achieve a very high discriminative accuracy - although the models never see the ground truth gender label while training.

Figure 13: First two principal components of the inferred embeddings, with the ground truth labels for gender and region of accent for the VCTK speakers as in Arik et al. [2017b].

# F Similarity scores

For the result in Table 3, Fig. 14 shows the distribution of the scores given by MTurk users as in [Wester et al., 2016]. For 10 sample count, the ratio of evaluations with the 'same speaker' rating exceeds 70 % for all models.

Figure 14: Distribution of similarity scores for 1 and 10 sample counts.

## Footnotes

[12]Enrollment audios are from the same speaker.