[Reviews · NeurIPS 2018]

Reviewer 1



This paper investigates cloning voices using limited speech data. To that end, two techniques are studied: speaker adaptation approach and speaker encoding approach. Extensive experiments have been carried out to show the performance of voice cloning and also analysis is conducted on speaker embedding vectors. The synthesized samples sounds OK, although not in very high quality given only a few audio samples. Below are my details comments. 1. The investigated approaches have been used in the ASR domain under the name of speaker adaptation for years. When GMM-HMM was dominant, speaker adaptation was typically conducted by linear transformations. Since DNN-HMM became the state of the art, using speaker embedding vectors is one of the popular approaches for speaker adaptation. Among various speaker embedding algorithms, i-vector is the one that is widely used but there are also other speaker embedding and encoding techniques. Some of the techniques are actually very similar to what is used in this paper. For instance, [1] Ossama Abdel-Hamid and Hui Jiang, "Fast speaker adaptation of hybrid NN/HMM model for speech recognition based on discriminative learning of speaker code," ICASSP 2013. [2] Xiaodong Cui, Vaibhava Goel, George Saon, "Embedding-based speaker adaptive training of deep neural networks," Interspeech 2017. From this perspective, the speaker adaptation techniques investigated in the paper are similar in spirit. That being said, ASR and TTS have very different lines of using speaker information, almost in the opposite directions. It is good to see those techniques can help the neural voice cloning. This usually didn't happen in the GMM-HMM domain. 2. In terms of speaker embedding, the argument of whether using a separate speaker encoder or a joint one is not clear from the paper. The authors claim that one of the drawbacks of I-vectors is to train a separate speaker encoder while the proposed framework can conduct the joint training. However, in the experiments, the observations on the joint training are that the estimated embedding vectors tend to smear across different speakers and a separate encoder has to be trained. This is a very common issue for joint training of speaker embedding with the main network. Given the nature of the problem, that is, using just a few samples to adapt a large network with substantial number of parameters, overfitting is the crucial issue. My sense is that there should be some way to actually isolate out the speaker embedding part to make it stand-alone. 3. I am confused by the cloning time for the speaker adaptation column in Table 1. Embedding-only takes about 8 hours with 128 parameters per speaker while the whole model takes half to 5 mins with 25M parameters. Is this a typo or I am missing something here? 4. It would be helpful to probably show the trend of performance using various amounts of cloning data. With only a few audio samples, the cloning may probably just pick up some very fundamental features of the speaker such as accent, pitch, gender, etc.. How would it perform in the given framework when presented large amounts of data? Will the performance keep improving or will it simply plateau at some point and never reach the stand-alone TTS performance? In ASR, people usually show the trend with various amounts of adaptation data. For some techniques when having a large amount of data they approach the speaker dependent system while for some other techniques they will never reach that level. I think this scenario worth investigating. 5. Instead of just showing the text to generate the audio samples, it would be more rigorous to show the audio samples in, say, seconds or minutes. At the end of the day, it is this metric that really matters.

Reviewer 2



The paper presents a system to clone voices, allowing one to build a speech synthesis system with a new voice provided just a few seconds of adaptation (or "encoding") data. This is an extremely relevant research topic, and the results achieved seem pretty good to me (although the quality of synthesised speech produced by research systems seems pretty low, IMHO). The main contributions of this work are a detailed analysis of the issues in speech synthesis, and the presentation of a new "encoding" based approach, which is more efficient. The examples also allow th reader to see for himself/ herself the quality of the provided speaker embedding method, which is also more efficient than existing techniques.

Reviewer 3



This work examines the problem of cloning a target speaker's voice given a trained generative model (with audio/speech and text). The broader context of the work is in Text to Speech (TTS) models in which rapid and excellent developments have occurred in the last few years. These models use DNNs as seq2seq models to predict speech to text. Other permutations are speech to speech/voice conversion (going from source voice to target voice), and voice cloning and adaptation, which this paper addresses. In the problem under consideration in the paper (voice cloning) one is able to produce voice that sounds like a certain 'target' or desired speaker using only a few samples presented which is then used to generate speech given arbitrary text. This generation is carried out using a multispeaker generative model that is trained over a large number of speakers and text/speech training data. The reviewer feels that this work is an outstanding example of current TTS work and a massive undertaking both in terms of the quality of results produced and in the relevance of ideas it contains. Prior work: The work is an extension of recent developments on seq2seq attention based TTS modeling works (e.g. Tacotron, DeepVoice-3 and lower). The models have several important improvements such as autoregressive models with dilated convolutions, trainable speaker embeddings, multiheaded self attention, and input feature processing (convolutions, filter banks etc from CBHG in tacotron), and using characters instead of phonemes to do away with segmentation. The work uses the DeepVoice-3 architecture to carry out the voice cloning task. Cloning approach: Two approaches are presented. 1) Speaker adaptation - Step1: A multispeaker generative model is trained first with large amount of data (text/speech). Step 2: In order to clone voice of a new speaker (with few samples), the model learns the embedding of the speaker using the trained generative model learnt with large amount of data. This is what they call the speaker adaptation approach without fine tuning. In addition to the above, one may also make the generative model tunable (along with the new speaker's embeddings). This is called 'whole model adaptation' in the paper. It is noted that the whole model adaptation produces the best speaker samples, and trains much faster than the case where only the speaker is implicitly adapted but the model remains fixed. Step 3: The adapted speaker embedding is now used for the inference task where text is passed along with speaker encoding to produce voice from the new speaker. 2) Speaker encoding - Step 1: The first step is to train the generative model along with speaker embeddings as in the speaker adaptation appraoch. The difference here is that the model also learns to produce as output an encoding for speakers. Step 2: For the voice cloning task, the model outputs a speaker encoding explicitly when presented with voice samples from the target to be cloned. Step 3: After an encoding/embedding is produced in the second step, one may use it for inference in the speaker adaptation approach where one passes in some text and the newly produced encoding to generate voice from the cloned speaker. The paper goes on to show results to demonstrate that both approaches are viable (samples are available at https://audiodemos.github.io/ as indicated in the paper). There are many details and minutiae in the work of very fine granularity. A few points came to the attention of this reviewer. 1) Training speaker encoding: Lines 128, 129 in Section 3.2 (Speaker encoding). Optimization challenges are mentioned while training the speaker encoding approach. The model tends to average out speaker voices to minimize overall generative loss. Instead, they device a workaround by training each speaker separately and fine tuning the model afterwards. 2) In addition to human evaluations, the architecture produces two comparison metrics to assist in the verification process. a) Speaker classification - determining which speaker an audio sample belongs to b) Speaker verification - verifying the identity of the speaker. In other words, performing binary classification to identify whether the test audio enrolled is from the same speaker. c) Latent space walks - While this work does not claim to explore stochastic (in the style of VAE) latent manifolds, it presents similar latent style transformations by manipulating embeddings (section 4.4 - Voice morphing via embedding manipulation). This could be useful pointers for investigations in tasks such as transforming voices. Results: A fairly extensive set of experiments are conducted to compare quality of generated samples. Notably, the quality improves with the number of samples used for cloning (the number of samples used varies from 1-100). Also, the results consistently show that whole model adaptation is 'better' (when we listen to the samples) than single speaker embedding. The speaker embedding and encoding performance seem to be on par. While I think that the paper's subject matter, ideas and results are truly exploring the limits of what is possible now in neural TTS, it seems that this would be an extremely difficult project to replicate owing to the complexity of the models presented and the generally pipelined nature of the space (it is not really end-to-end). Comments on pertinent evaluation criteria: 1) Quality: Excellent. Encyclopedic presentation of current neural TTS landscape. Several novel ideas. 2) Clarity: Excellent. The paper is readable and well written. 3) Originality: Very Good. Voice cloning is an interesting (although these problems have been around for a long time) application for DNNs. The paper presents several original ideas in speaker adaptation, encoding and latent vector manipulation. 4) Significance: Excellent. The problems explored (text to speech, voice cloning, voice conversion, speaker verification etc.) in this paper are very pertinent to speech synthesis research. It demonstrates the massive progress made in recent times in DNN based TTS.